# Early Life Stress Is Associated with Alterations in Lymphocyte Subsets Independent of Increased Inflammation in Adolescents

**DOI:** 10.3390/biom14030262

**Published:** 2024-02-22

**Authors:** Brie M. Reid, Christopher Desjardins, Bharat Thyagarajan, Michael A. Linden, Megan Gunnar

**Affiliations:** 1Department of Psychiatry and Human Behavior, Warren Alpert Medical School, Brown University, Providence, RI 02912, USA; 2Saint Michael’s College, Colchester, VT 05439, USA; cdesjardins@smcvt.edu; 3Department of Laboratory Medicine and Pathology, University of Minnesota, Minneapolis, MN 55455, USA; thya0003@umn.edu (B.T.); linde013@umn.edu (M.A.L.); 4Institute of Child Development, University of Minnesota, Minneapolis, MN 55455, USA; gunnar@umn.edu

**Keywords:** early life stress, inflammation, T cells, B cells, senescence, adolescence

## Abstract

Early life stress (ELS) is linked to an elevated risk of poor health and early mortality, with emerging evidence pointing to the pivotal role of the immune system in long-term health outcomes. While recent research has focused on the impact of ELS on inflammation, this study examined the impact of ELS on immune function, including CMV seropositivity, inflammatory cytokines, and lymphocyte cell subsets in an adolescent cohort. This study used data from the Early Life Stress and Cardiometabolic Health in Adolescence Study (N = 191, aged 12 to 21 years, N = 95 exposed to ELS). We employed multiple regression to investigate the association between ELS, characterized by early institutional care, cytomegalovirus (CMV) seropositivity (determined by chemiluminescent immunoassay), inflammation (CRP, IL-6, and TNF-a determined by ELISA), and twenty-one immune cell subsets characterized by flow cytometry (sixteen T cell subsets and five B cell subsets). Results reveal a significant association between ELS and lymphocytes that was independent of the association between ELS and inflammation: ELS was associated with increased effector memory helper T cells, effector memory cytotoxic T cells, senescent T cells, senescent B cells, and IgD− memory B cells compared to non-adopted youth. ELS was also associated with reduced percentages of helper T cells and naive cytotoxic T cells. Exploratory analyses found that the association between ELS and fewer helper T cells and increased cytotoxic T cells remained even in cytomegalovirus (CMV) seronegative youth. These findings suggest that ELS is associated with cell subsets that are linked to early mortality risk in older populations and markers of replicative senescence, separate from inflammation, in adolescents.

## 1. Introduction

Exposure to adverse conditions early in life (early life stress, ELS) is associated with an increased risk of poor health and early mortality [1,2]. ELS encompasses a wide range of adverse early-life experiences, including parental separation, neglect and deprivation, childhood abuse, and low socioeconomic status (SES). Worldwide, the prevalence of ELS is high [3] and impacts children across countries and income levels [4]. ELS carries consequences for cognitive, social, and neurological development [5], and at its most extreme is associated with stunted growth and chronic stress [6]. Although the specific pathophysiological effects of ELS vary, the immune system is implicated in long-term outcomes after ELS exposure, including chronic inflammatory disorders like autoimmune diseases, allergies, and asthma [7]. Recent research has focused on understanding how ELS affects immune function, including inflammation, impairment of the cellular immune system, and immunosenescence [8].

Research on the relationship between ELS and immune functioning in human cohorts has examined inflammation in aging [9], as aging is accompanied by a persistent, low-grade inflammatory state referred to as “inflammaging” [10,11,12]. This chronic inflammation can result in cellular damage and is implicated in the onset and progression of various age-related conditions, including cardiovascular disease [13,14] and neurodegenerative disorders [15]. One robust finding in studies of adults with a history of ELS is the association with elevated levels of typical inflammation markers, such as white blood cell count, circulating proinflammatory cytokines, and C-reactive protein (CRP) [16,17]. However, studies looking at the developmental sequelae of ELS on the immune system have found inconsistent evidence for the link between ELS and inflammation in children and adolescents [18]. Despite the growing interest in the relationship between ELS and developing a proinflammatory phenotype, a meta-analysis of children and adolescents exposed to ELS found relatively limited evidence for such associations [19]. To understand the inconsistent association between ELS and changes to the immune system in younger populations, recent research has begun to examine additional aspects of immunity following exposure to ELS.

ELS has been linked to indicators of accelerated aging across various systems [20,21,22]. Notably, there is a growing focus on understanding how ELS affects long-term health by accelerating the aging process in the adaptive immune system, known as immunosenescence [8]. The adaptive immune response involves antigen-specific B and T lymphocytes and the development of immunological memory. Immune senescence is a complex process marked by changes in the immune system that may accelerate biological decline and the onset of chronic diseases [23]. Changes in the adaptive immune system contribute to an aging-related immune profile characterized by lymphocyte subsets that eventually diminish the ability to effectively respond to new infections, vaccinations, and antigens [24,25]. This includes a decline in naive T cells, an accumulation of memory T cells, reduced antibody production, and decreased natural killer cell cytotoxicity [26,27,28].

Previous studies have found links between immune senescence and early mortality, though most have focused on older adults and hospitalized patients. In the Health and Retirement Study (HRS), low CD4+ naive T cell counts were associated with a higher risk of aging-related outcomes, such as disability, multiple health conditions, and mortality [29]. In the same cohort, CD4+ effector memory T cells and IgD− memory B cells were nominally associated with increased mortality odds [30]. Among older adults [31], a higher count of CD8+ memory T cells was associated with increased mortality, even after accounting for chronological age. Similarly, sepsis patients with a low percentage of total lymphocytes exhibited higher overall mortality rates [32]. In hemodialysis patients, one study found that depletion of naive T cells and an increased percentage of CD8+ central-memory T cells were associated with an elevated risk of all-cause mortality [33], while another study showed that lower B cell counts were linked to higher odds of all-cause mortality [34]. In a retrospective study of COVID-19 patients, all-cause mortality was associated with lower levels of lymphocytes and NK cells and higher levels of CRP [35]. These findings suggest that an aging immune system is associated with short-term mortality, independently of age-related inflammation or other physiological dysfunction measures.

### Immune Senescence and Early Life Stress

Early rodent models found acute and chronic effects of ELS on immunity, including compromised immune competence and altered response regulation [36,37,38,39]. In nonhuman primate models, prolonged maternal deprivation in the form of nursery rearing during infancy has been used as a model of ELS and found immune suppression and altered immune responses [40] as well as reduced survival and immune alterations in rhesus monkeys [41]. These studies strongly suggest that early stress conditions can leave lasting impacts on the immune system [42]. Studies in humans have extended this work to understand the effects of early adversity on cellular aging and immune function. To investigate whether stress experienced in very early life impacts long-term immune function, studies have turned to children and adolescents with a history of institutionalization during infancy. Institutionalization in infancy is the human analog of non-human primate nursery rearing and is a well-studied form of ELS, characterized by deprivation of species-expectant experiences for young humans, including deprivation of basic needs like proper healthcare and nutrition, inadequate cognitive and social stimulation, and a lack of stable, consistent adult attachment figure [6]. Current studies on previously institutionalized (PI) children focus on those who typically experienced institutional care as infants, which ended when they were adopted into well-resourced families, most often before the age of 3. Importantly, studying PI children enables researchers to isolate the specific impact of early-life stress on later immune system development while reducing the likelihood of ongoing impairments in nurturing care.

The inconsistent association between ELS and an inflammatory phenotype is replicated in studies on early institutional deprivation. Previously institutionalized Romanian adolescents did not exhibit elevated circulating inflammatory factors (e.g., CRP, IL-6) [43]. Similarly, a separate study of PI adults found no increases in circulating proinflammatory cytokines or larger increases in these cytokines following in vitro stimulation with various antigens [44]. Research conducted by our team revealed that previously institutionalized (PI) youth exhibited higher circulating levels of Tumor Necrosis Factor-alpha (TNFα), but not Interleukin-1β (IL-1β) or Interleukin-6 (IL-6) [45]. Recent research has honed in on the immune cell profiles of children who experienced early institutional care, particularly T cell subsets, which are believed to be sensitive to early experiences. Two studies have observed an increase in CD8+ T cells relative to CD4+ T cells in PI youth compared to their counterparts, which, when extreme, indicates immune incompetence [46,47]. These findings align with evidence that PI youth struggle to control viruses like Epstein–Barr, herpes simplex, and cytomegalovirus (CMV) [48]. Additionally, two studies have reported higher levels of immune senescence markers (CD57) on CD4+ and CD8+ T cells in PI adolescents and adults relative to their non-adopted peers [47,49]. Additionally, the percentage of senescent CD8 T cells mediated the heightened inflammatory response observed in PI individuals [45]. Notably, the results from all of these studies suggest an important role for CMV seropositivity in PI youth and adults [47,49], raising the question of whether exposure to ELS is associated with changes in immune cell subsets without environmental exposure to CMV.

While these initial findings suggest that enduring immune differences persist years after leaving institutional care, comprehensive investigations into a broader range of lymphocyte subsets are lacking. Furthermore, no studies have explored whether the effects of rearing on immune senescence are distinct from those of inflammation. This knowledge gap hinders our understanding of the connections between ELS, accelerated aging, and immune dysfunction. To address this, our study focuses on adolescents aged 12 to 21 years who were internationally adopted from orphanages during infancy and toddlerhood. These individuals were raised in well-resourced households by educated parents in low-risk neighborhoods with access to quality healthcare. We compared them to a control group of similar age and sex raised developing from conception in comparable family environments. Our investigation specifically examines the impact of ELS on immune function, including CMV seropositivity, inflammatory cytokines, and lymphocyte cell subsets. We aim to ascertain whether the effects of ELS extend to inflammatory cytokines, B cells, and T cells, and we hypothesize that ELS will be associated with changes in lymphocyte subsets independent of inflammation. We further conducted exploratory analyses in a subset of CMV-negative participants to understand whether associations between ELS and immune phenotype are independent of CMV exposure. 

## 2. Materials and Methods

### 2.1. Participants

Participants were from Wave 1 of the longitudinal Early Life Stress and Cardiometabolic Health in Adolescence Study (CardioHealth Study). Ninety-seven PI youth who were adopted internationally from orphanage-like institutions and ninety-six youth born and raised by birth families (non-adopted, NA) were recruited from Minneapolis, Saint Paul, and nearby areas. Youth participants were included if they were 12–21 years of age and they or their parents, in the case of college students, currently reside within driving distance of the University of Minnesota’s Clinical and Translational Science Institute (CTSI) where the study was conducted. For PI youth, inclusion requires a known age of adoption and at least 50% preadoption life in institutional care. Youth participants were excluded if the parents reported that youth were born prematurely (less than 37 weeks of the gestational age), had congenital and/or chromosomal disorders (e.g., cerebral palsy, FAS, intellectual disability, Turner Syndrome, Down Syndrome, Fragile X), autism spectrum disorders, history of serious medical illness (e.g., cancer, organ transplant), if they were taking systemic glucocorticoids, or if they were taking beta-adrenergic medications and could not go without medication for 24–48 h before testing.

The University of Minnesota, Twin Cities Institutional Review Boards approved all procedures. All minor participants assented and their parents provided informed consent, and participants who reached the age of majority provided consent.

### 2.2. Protocols and Measures

Data collection involved two sessions at Wave 1. During session 1, conducted virtually, youth participants and their parents completed online questionnaires on their demographics, health, and life experiences. In session 2, conducted early in the morning, fasted youth participants and their parents attended a clinic visit at the CTSI. At the clinic, participants underwent comprehensive health checks, which involved the collection of immune and metabolic biomarkers and anthropometrics. Early life stress was operationalized as a dichotomous group, PI vs. NA. 

Fasting blood (>8 h) samples were collected to assess CMV seropositivity, High Sensitivity C-reactive protein (hsCRP), Interleukin-6 (IL-6), tumor necrosis factor alpha (TNFa), and T cell immunophenotyping. Blood was drawn by a trained nurse and processed within 72 h. 

#### 2.2.1. Immune and Inflammatory Measures

Samples were stored at −80 °C and subsequently tested by the Cytokine Reference Laboratory (CRL, CLIA’88 license #24D0931212) at the University of Minnesota. Samples were analyzed for Human C-Reactive Protein (CRP) using a colorimetric ELISA kit (R&D Systems, Minneapolis, MN, USA; cat # DCRP00B) according to the manufacturer’s instructions and read on a Synergy XL plate reader. Samples were tested in duplicate and values were interpolated from standard curves of recombinant protein. Samples were analyzed for human-specific IL-6 and TNFa as a 3-plex using the Ella microfluidic platform with Simple Plex Carts (Protein Simple, Wallingford, CT, USA; cat. # SPCKC-PS-003413). The assays were performed according to manufacturers’ instructions and the averages of triplicate values were interpolated from standard curves specifically calibrated by the manufacturer for the cartridges used.

CMV antibody status was determined with quantitative measurement of IgG antibodies to CMV to indicate past or current infection using chemiluminescent immunoassay (M Health Fairview Reference Laboratory). Values of 0.70 U/mL or greater suggest previous exposure or immunization and probable immunity and were subsequently categorized as seropositive for CMV.

##### Immunophenotyping

Heparinized blood was delivered immediately to the Advanced Research and Diagnostics Laboratory at the University of Minnesota Dept. of Lab Medicine for immunophenotypic characterization. Immunophenotyping was performed on whole blood samples stained after red blood cell (RBC) lysis by alkali. For lymphocyte subset analysis, lymphocytes were selected based on forward versus side scatter (FSC vs. SSC) plots. Antibodies were purchased from either BD Biosciences (San Jose, CA, USA) or BioLegend (San Diego, CA, USA). Samples were collected and analyzed on a Fortessa X20 flow cytometer (BD Biosciences, San Jose, CA, USA). The flow cytometric data were collected with FACSDiva (BD Biosciences, San Jose, CA, USA) and analyzed with FloJo (Ashland, OR, USA).

The markers used to identify 21 immune cell subsets can be found in Appendix A. These immune cell subsets were presented as percentages relative to their parent population. Specifically, total T cells, NK T cells, and B cells were represented as percentages of the total lymphocytes. The CD4+ and CD8+ subsets were expressed as a percentage of the total T cells. Additionally, subsets of CD4+ and CD8+ cells, such as effector, effector memory, central memory, and naive cells, were expressed as percentages of the CD4+ and CD8+ cell populations, respectively. Similarly, subsets of B cells were expressed as percentages relative to the total B cell population.

##### Gating Strategy

The gating strategies employed in this analysis are visually presented in Appendix A.

In the flow cytometry panels used to measure 21 immune cell subsets, the following approach was taken for the lymphocyte panel. Lymphocytes were initially identified based on their appearance in an FSC-A/SSC-A dot plot. Among the identified lymphocytes from the FSC-A/SSC-A dot plot, single cells were selected using an FSC-W/FSC-H dot plot. Live, single lymphocytes were further distinguished based on their viability dye/SSC-A dot plot. T cells (CD3+ CD19−) and B cells (CD3−CD19+) were then separated using a CD3/CD19 dot plot (Appendix A). Among the B cells, further division into three subsets was accomplished using an IgD/CD27 dot plot: IgD+ memory B cells (CD3−CD19+CD27+IgD+), IgD− memory B cells (CD3−CD19+CD27+IgD−), and naive B cells (CD3−CD19+CD27−) (Appendix A). In addition, senescent B cells (CD3-CD19+CD57+) were also identified using a dot plot (Appendix A). T cells, such as cytotoxic T cells (CD3+ CD4−CD8+) and helper T cells (CD3+ CD4+CD8−), were achieved using a CD4/CD8 dot plot, and NK T cells (CD3+CD56+) and senescent T cells (CD3+CD57+) were accomplished using dot plots, as shown in Appendix A. In addition, both cytotoxic and helper T cells were then categorized into four subsets based on a CCR7/CD45RA dot plot: effector (EFF, CD45RA+, CCR7−), effector memory (EM, CD45RA−, CCR7−), central memory (CM, CD45RA−, CCR7+), and naive (N, CD45RA+, CCR7+) cytotoxic or helper T cell subsets (Appendix A). The helper and cytotoxic T cell subsets were also categorized into senescent helper and cytotoxic T cell subsets using the CD4+/CD57+ dot plot and the CD8+/CD57+ dot plot, respectively (Appendix A).

#### 2.2.2. Covariates

Anthropometrics. During the clinic visit, the participant’s height (in cm) and weight (in kg) were assessed by a nurse and a trained student researcher. Height and weight were assessed to the nearest 0.1 cm and the nearest 0.1 kg, respectively, using a wall-mounted stadiometer and electronic scale and were used to calculate body mass index (BMI). All female participants were screened for a negative pregnancy test before undergoing a DXA scan. Total body composition was obtained using standard procedure in the supine position on a GE Lunar iDXA (iDXA; General Electric Medical Systems, Madison, WI, USA) and scans were measured using enCore^TM^ software (platform version, 16.0, General Electric Medical Systems).

Demographics. Parents completed information on family demographics, including their child’s age, parent education, family pre-tax income, and number of individuals in the household. Youth also self-reported participant demographics and age. All self-reported data were collected via online questionnaires in REDcap. Youth age (years) and sex assigned at birth were used as covariates in all analyses. 

Time at blood draw. Time at blood draw (minutes since midnight) was used as a covariate in all analyses.

### 2.3. Statistical Approach

All analyses were performed using R (version 4.3.2). Prior to analysis, data were preprocessed to correct for measurement errors. Q-Q plots were visualized to assess normality and considered transformations of the response variables that best met normality and homogeneity of variance. This prepossessing involved winsorizing extreme outliers (observations > 6 SD away from the mean) to the nearest plausible values (i.e., the next value not deemed an extreme outlier). This was performed for IL-6 (one observation), TNFA (one observation), senescent B cells (two observations), effector helper T cells (two observations), and central memory cytotoxic T cells (one observation).

#### Analytical Approach

Three analytical approaches were used. First, to examine group differences in CMV, a chi-square test of association was performed. Second, to examine the dimensionality of IL6, CRP, and TNFA, we performed a principal components analysis (PCA) after we standardized (Z-score normalization) these variables and extracted the first principal component (PC). Next, we regressed the first PC onto group, sex, age, total tissue percent fat, and time of blood draw. Third, to compare the impact of early life stress on cell subsets, we regressed each cell subset type onto the group, sex, age, time of blood draw, IL6, and CRP. To adjust for an inflated type I error rate due to multiple comparisons, we used the Benjamini–Hochberg correction [50]. For the last two sets of analysis, the response variable was transformed as needed to meet modeling assumptions of homogeneity of variance and normality of residuals. For T cells, no adequate transformation was found, therefore, a permutation test was performed. Relative risk (RR) and Cohen’s *f*^2^ were used as effect size measures [51].

## 3. Results

We employed regressions controlling for sex, age, total fat percentage, and time of blood draw to investigate the influence of early life stress (ELS) on inflammatory cytokines, B cells, and T cells, and whether ELS-related changes in lymphocyte subsets are independent of inflammation. Demographics of the participants are reported in Table 1. The mean age that PI youth were adopted from institutional care was 16.1 months, and at the time of the blood draw, the PI youth were slightly older than non-adopted youth. The adolescents from both rearing conditions were raised by families in the Upper Midwest of the United States. Primary caregivers self-reported their race and ethnicity as predominately white (91.6%) and not Hispanic or Latinx (98.9%). The median annual household income for families was USD 100,001 to USD 150,000 per year and ranged from USD 25,000 to >USD 200,000 per year. The groups did not differ based on primary caregiver educational attainment, total household income, total adipose tissue, known allergies, or time of blood draw.

### 3.1. Group Differences in CMV

There was a significant association between the groups and the presence of the CMV antibody ($\chi^{2}=48.857,\;df=1,\;p<0.001$). The proportion of PI youth that were seropositive for CMV was 73% compared to 27% of NA youth (RR = 2.75).

### 3.2. Dimensionality of IL6, CRP, and TNF-a

The first PC explained 54% of the variability in IL6, CRP, and TNFA. All loadings were positive and medium/large in strength (IL6 and CRP were 0.63 and TNFA was 0.45, respectively). Boxplots of the first PC by group are shown in Figure 1. There was a significant difference in the groups controlling for sex, age, total tissue percent fat, and time of blood draw (t=2.14, df=166, p=0.03), with PI youth expected to be 0.36 higher on the first PC than NA youth (Cohen’s $f^{2}=0.06$) (Figure 1).

### 3.3. Lymphocyte Subset Analyses

Appendix A presents the surface markers of cell types and Appendix A presents the non-transformed means and distributions of cell types. The complete results of the cell subset analysis are presented in Table 2, and full model output with covariates and un-adjusted *p*-values are presented in Appendix A.

### 3.4. B Cells

For the B cells, only IgD− memory B cells and senescent B cells differed by group (Figure 2). For IgD− memory B cells, PI youth were expected to have more IgD− memory B cells than NA youth (Est=0.33, t=2.71, df=160, BH corrected p=0.025, Cohen’s $f^{2}=0.07$). For senescent B cells, PI youth were expected to have more senescent B cells than NA youth (Est.=0.08, t=3.088, df=160, BH corrected p=0.013, Cohen’s $f^{2}=0.05$). 

### 3.5. T Cells

#### 3.5.1. Helper T Cells

For the T cells, helper T cells (Est=−3.7, t=−2.90, df=160, BH corrected p=0.018, Cohen’s $f^{2}=0.06$), effector memory T cells (Est=4.7, t=3.40, df=160, BH corrected p=0.009, Cohen’s $f^{2}=0.19$) differed by group (Figure 3). The group difference in naïve helper T cells (Est=−4.5, t=−2.36, df=160, p=0.019, BH corrected *p* = 0.051 Cohen’s $f^{2}=0.11$) did not survive multiple comparison corrections. For both helper T cells and naïve helper T cells, PI youth were expected to have a lower number of helper T cells and naïve helper T cells than NA youth, while PI youth were expected to have more effector memory T cells than NA youth.

#### 3.5.2. Cytotoxic T Cells

For the cytotoxic T cells, effector memory cytotoxic T cells and naïve cytotoxic T cells differed by group (Figure 4). For effector memory cytotoxic T cells, PI youth were expected to have more effector memory cytotoxic T cells than NA youth (Est=3.0, t=3.26, df=160, BH corrected p=0.009, Cohen’s $f^{2}=0.09$). For naïve cytotoxic T cells, PI youth were expected to have fewer naïve cytotoxic T cells than NA youth (Est=−6.9, t=−2.6, df=160, BH corrected p=0.032, Cohen’s $f^{2}=0.07$). The group difference in cytotoxic T cells, where PI had more cytotoxic T cells than NA youth (Est=2.6, t=2.61, df=160, p=0.03, *BH corrected p* = 0.07 Cohen’s $f^{2}=0.04$), did not survive multiple comparison corrections.

#### 3.5.3. Regulatory T Cells

The regulatory T cells did not differ by group (Est=0.1, t=1.52, df=160, BH corrected p=0.178, Cohen’s $f^{2}=0.02$).

#### 3.5.4. Senescent T Cells

For the senescent T cells, only senescent T cells differed by group: PI youth were expected to have more senescent T cells than NA youth (Est=0.46, t=4.94, df=160, *BH corrected p* =< 0.001 Cohen’s $f^{2}=0.19$) (Figure 5). PI youth were also expected to have more senescent helper T cells than NA youth (Est=3.2, t=2.11, df=160, p=0.037, *BH corrected p* = 0.077 Cohen’s *f*^2^ = 0.03), though this difference did not survive correction for multiple comparisons. 

### 3.6. Natural Killer T Cells

Finally, the natural killer T cells did not differ by group (t=1.50, df=160, p=0.178, Cohen’s $f^{2}=0.04$).

### 3.7. Sensitivity Analyses

We performed two sensitivity analyses. The first sensitivity analysis examined whether the findings differed when we controlled for the first principal component instead of IL-6 and CRP. The results were qualitatively the same (i.e., no results changed from significant to non-significant or vice versa) and the largest change in Cohen’s $f^{2}$ was 0.003 for effector memory helper T cells. Thus, the results using IL-6 and CRP are presented for ease of interpretation.

The second sensitivity analysis was an exploratory analysis where we compared the two groups using only participants that were CMV negative. All models were rerun on these 64 participants (NA =59, PI = 15). Due to the reduced power and exploratory nature of this analysis, we did not control for multiple comparisons. The groups differed such that PI youth without CMV had fewer IgD+ memory B cells (Est = −0.363, t=−2.14, df=60, p=0.037, Cohen’s $f^{2}=0.06$), fewer helper T cells (Est = −7.05, t=−2.71, df=60, p=0.009 Cohen’s $f^{2}=0.15$), more cytotoxic T cells (Est = 5.18, t=2.30, df=60, p=0.025, Cohen’s $f^{2}=0.10$), and more natural killer T cells (Est = 3.25, *t =*
2.07, df=60, p=0.043, Cohen’s $f^{2}=0.12$) compared to NA youth without CMV. Thus, while the direction of the differences between the groups remained between the main analyses and the exploratory CMV-negative analyses, the only associations that were consistent between the main analyses and the exploratory analyses were the finding that PI youth had fewer helper T cells and more cytotoxic T cells (before correcting for multiple comparisons). Appendix A displays the results for the association between early life stress exposure and cell subsets in CMV-negative PI and NA youth.

## 4. Discussion

Early life stress (ELS) poses a risk to long-term health and mortality, with growing evidence underscoring the immune system’s pivotal role in shaping these outcomes. Our study examined how ELS in the form of institutional care in infancy was associated with inflammatory markers (IL6, CRP, and TNF-a) and lymphocyte subsets in healthy adolescents. The results show that ELS is associated with distinctive lymphocyte profiles and heightened inflammation in adolescents and young adults, decades after removal from institutional care into well-resourced homes. We observed a positive association between ELS exposure and inflammation, as indicated by a principal component derived from IL6, CRP, and TNF-a. Furthermore, we found that PI youth exhibited more IgD− memory B cells and senescent B cells. PI youth had a lower percentage of helper T cells but higher levels of effector memory T cells. Additionally, effector memory cytotoxic T cells were more abundant, while naive cytotoxic T cells were diminished in PI youth. Senescent T cells were also increased in the PI group. In exploratory analyses of youth who were CMV negative, only the association between ELS and fewer helper T cells and increased cytotoxic T cells remained.

These differences suggest that the PI group’s immune system has undergone more extensive pathogen exposure and maturation, leading to a state of advanced immunological aging compared to that of non-adoptive counterparts. These differences also suggest that CMV exposure may not be the only link between ELS and changes in immune phenotype. The cohort studied consists of clinically healthy adolescents, with three-quarters of the sample reporting no known allergies. Our study, encompassing the largest cohort examining lymphocyte subsets and inflammation in PI youth to date, with double the sample size of previous research, underscores that immune cell subsets remain independently associated with ELS even after adjusting for inflammation. This suggests that ELS may also influence the immune system through mechanisms unrelated to systemic chronic inflammation, highlighting the intricate interplay between replicative senescence of immune cells and cytokine secretion. These findings offer insights into potential immune-related health outcomes in ELS-exposed individuals.

### 4.1. Inflammation

PI youth exhibited higher levels of inflammation, as indicated by a principal component composed of IL6, CRP, and TNF-a. This finding contrasts with our previous observations in PI youth, where we found no significant differences in peripheral markers of inflammation (e.g., leukocyte counts, CRP, IL-1β, IL-6) when compared to non-adopted (NA) youth [45,47]. Previous research, including meta-analyses, presents a complex picture of the relationship between early life adversity and inflammation, showing small or non-significant effects in younger populations [19,47] but significant associations in adults [17]. This implies that developmental factors may influence the immune response to early life stressors, yet further investigation is essential to pinpoint these factors responsible for the observed variations. 

One explanation for the disparity between childhood and adulthood findings could be that dysregulated immune function, characterized by elevated inflammatory markers, may not manifest until later in human development. An alternative hypothesis centers on alterations in the developing hypothalamic–pituitary–adrenocortical (HPA) axis as a potential driver of these developmental differences in inflammation response. The HPA axis regulates cortisol release, a primary stress hormone governing the human stress response. In adults, chronic stress is associated with an increased release of inflammatory cytokines and heightened inflammatory responses, which are linked to various health disorders [52]. In humans and non-human primates, early-life deprivation often results in a shift towards HPA hypo-functioning (blunted cortisol response), a phenomenon observed in PI children [6]. However, our prior research in a separate cohort of PI children and adolescents has provided initial evidence that puberty may offer a window of plasticity during which the HPA axis can recalibrate away from hypo-cortisolism if current life conditions become less harsh. This recalibration increases HPA reactivity during adolescence, resulting in cortisol levels in PI youth similar to their peers without adverse histories [53]. Consequently, if the HPA axis plays a pivotal role in inflammatory stress response, a hypo-responsive HPA axis in childhood may serve as a buffer against a pro-inflammatory phenotype, safeguarding individuals from heightened inflammation. If the HPA axis shifts from hypo- to hyper-reactivity during the transition to adulthood, excessive cortisol levels may stimulate the synthesis and release of excess inflammatory cytokines within the immune system [54]. Thus, adolescents and young adults experiencing this “recalibration” could develop glucocorticoid-resistant immune cells, resulting in observable changes in basal cortisol levels and inflammation. This alternative hypothesis suggests that PI youth may initially have a less responsive HPA axis during early adolescence, which later becomes overactive as they age, potentially contributing to a pro-inflammatory state and providing insight into the HPA axis’s role in the connection between early life stress and inflammation-related health disorders in adulthood. We note that this hypothesis is speculative, and further research is warranted to elucidate the HPA axis’s precise role in the development of inflammation-related mental and physical health disorders.

### 4.2. Cell Subsets

#### 4.2.1. B Lymphocytes

PI youth displayed higher levels of IgD− memory B cells and senescent B cells compared to NA youth. B cells play a crucial role in adaptive immunity by producing specific antibodies in response to antigens [27]. Memory B cells, such as IgD− memory B cells, are known to increase with age, contributing to a decline in immune diversity and a reduced capacity to respond to new antigens [27]. In previous research, IgD− memory B cells have been associated with increased mortality odds [30], and memory B cells have been linked to cardiovascular mortality [55]. Memory lymphocytes are critical for immune responses, as they facilitate recall responses to previously encountered antigens and are important for effective vaccination [11,26].

#### 4.2.2. T Lymphocytes

PI youth had fewer CD4+ helper T cells but more effector memory T cells. These findings align with previous research showing a decrease in CD4+ cell percentages in PI youth and highlight differences in helper T cell subsets compared to typical aging patterns [46,56,57]. Similarly, PI youth displayed more effector memory cytotoxic T cells but fewer naive cytotoxic T cells. CD8+ memory T cells tend to increase with age [56], contrasting with stable CD8+ naive cells over childhood [56]. Cytotoxic T cells are crucial for host defense against viral infections, and a decrease in naive T cells has been associated with elevated mortality risk [33].

Senescent T cells, characterized by CD57+ expression, were also more abundant in PI youth. This finding partially replicates prior research demonstrating increased senescent T cells in both CD4+ and CD8+ cells in PI youth [47]. In contrast to our previous study, we did not observe a significant increase in CD4+ CD57+ cells in this study. Further, the elevation in CD8+ CD57+ cells was only nominally associated with ELS exposure after multiple comparison corrections. These T cell differences were evident by early to mid-adolescence, consistent with findings in young adults adopted by three months of age [49]. CD57 serves as a marker for senescent T cells, with CD8+ T cells showing faster senescence than CD4+ T cells. Chronic viral infections can accelerate T cell senescence through telomere attrition, DNA damage, and inflammatory stress signals [58]. Senescent T cells exhibit a proinflammatory phenotype, contributing to inflammaging and increasing susceptibility to age-related disorders such as cardiovascular and metabolic diseases and neurodegenerative conditions [29,58].

T cells are pivotal in maintaining health and preventing disease, primarily comprising CD4+ and CD8+ T cell subsets. These T cells differentiate into various effector and memory cells upon antigen stimulation, contributing to immune responses and long-term protection [58]. Thymic involution is a well-documented change during aging, beginning in childhood and peaking around puberty [59]. Thymic involution results in the reduced generation of naive T cells, compensatory memory T cell expansion, and decreased peripheral T cell repertoire diversity, impairing pathogen detection. Stress, infection, obesity, pregnancy, and antineoplastic therapies can accelerate thymic involution [59]. Consequently, persistent antigen stimulation, inflammation, and thymic involution lead to an enrichment of memory T cells and a decline in naive T cells, particularly in the CD8+ subset. This skewed memory phenotype compromises responses to new antigens, posing challenges for developing effective vaccines for older individuals [60,61,62]. To that end, experimentally manipulated accelerated T cell aging in murine models results in T cell senescence and systemic aging features, affecting metabolic, musculoskeletal, cardiovascular, and cognitive health [58,59]. Our results underscore the complex dynamics of T cell subsets in the context of ELS.

#### 4.2.3. NK Cells

PI youth did not show any differences in NK cells compared to NA youth. This is slightly in contrast to recent research in a separate cohort of PI adolescents, which showed that adolescents exposed to ELS exhibited lower percentages of CD56 bright NK cells in circulation, higher TNF-a levels, and a greater likelihood of CMV infection [63]. It is possible that to see an effect of ELS on NK cells, an interrogation of additional NK lineages than examined in this present study is necessary.

#### 4.2.4. Cytomegalovirus (CMV) Exposure

The precise biological mechanisms underpinning our findings remain unclear. However, it is plausible that they partially result from increased CMV exposure in the institutional setting, as group care is a recognized risk factor for CMV and other pathogen exposure in young children. It is most reasonable to conclude that these findings are influenced by early viral exposure and sustained antigen exposure, rather than the absence of parenting and adversity severity leading to prolonged CMV infection with recurrent reactivation. Over two-thirds of PI youth were CMV-seropositive, compared to less than one-third of NA youth, a rate significantly higher than typical CMV seropositivity in the U.S. population [64,65]. In two studies of PI adolescents and young adults, institutional care was associated with increased CMV antibody titers, and CMV mediated the link between early institutional care and later T cell profiles [47,49]. Recent research in a separate cohort also found that PI adolescents who were latent carriers of CMV had an increased expression of NKG2C and CD57 surface markers on NK cells, including CD56 dim lineages [63]. Given the known impact of CMV on immune senescence, this viral exposure is a likely driver of our findings.

In an exploratory investigation focusing exclusively on participants seronegative for CMV, notable differences emerged between PI and NA youths. Specifically, PI youths exhibited a reduced count of IgD+ memory B cells and helper T cells, alongside an elevation in cytotoxic T cells and natural killer T cells, in contrast to their NA counterparts. This finding diverges from our primary analysis, which did not establish a relationship between exposure to early life stress (ELS) and the levels of IgD+ memory B cells or natural killer T cells. However, the trends observed for other cellular distinctions persisted when comparing the primary analysis with the CMV-negative exploratory analysis. 

The immune profiles observed in the main analyses of institutionalized children suggest that PI youth appear to be influenced by their exposure to infectious agents, potentially elucidating the observed patterns of diminished immune functionality and heightened inflammation within this demographic. Our data hint at a significant role for herpes viruses, specifically CMV, in driving these immune system alterations. If our interpretation holds true, it suggests that targeted interventions—like reducing overcrowding and enhancing hygiene standards in institutionalized settings—might mitigate these adverse immune responses. While the practical implementation of such measures warrants consideration, their potential impact cannot be overstated. This perspective also implies a relative downscaling of other ELS factors, like social deprivation, in contributing to immune dysfunction, although their influence on other developmental outcomes remains noteworthy. 

In contrast, the reduction in helper T cells and the increase in cytotoxic T cells within PI youths were consistent findings across both the main and exploratory analyses. This consistency underscores the potential significance of these immunological alterations in relation to early life stress. In this finding, it is critical to consider the parallel findings from primate studies. These studies, presenting similar immune response patterns, were conducted in environments not characterized by heightened exposure to CMV or other pathogens. Nursery-reared monkeys were maintained in relatively antiseptic conditions. This contrasts markedly with the natural and less sanitary setting where monkey mothers and troops typically reside. This discrepancy underscores the complexity of extrapolating findings across species and environments, and it necessitates a cautious interpretation of the role of CMV and similar pathogens in shaping immune responses, particularly in more controlled or antiseptic rearing conditions. As the sample size for this exploratory analysis was small, future research is warranted. These nuanced findings emphasize the multifaceted nature of ELS and its varied impact on child development, urging a tailored approach in addressing the unique challenges faced by children exposed to ELS. 

### 4.3. Limitations

While our findings are significant, several limitations warrant acknowledgment. First, despite all adoptees experiencing postnatal institutional care, we had no information about prenatal conditions. In addition, institutions differ in quality, although nearly all fail to sustain species-typical rates of physical and mental development. Future studies should strive to account for both prenatal and postnatal conditions, recognizing that maternal stress and poor nutrition during pregnancy, along with postnatal ELS, can collectively affect neurological and immunological development [66,67,68]. Maternal and fetal factors involved in metabolic programming may explain or influence the conclusions drawn from the current results. Such studies might not be able to be conducted with PI youth as, for many, there is simply no information about prenatal care. 

An additional limitation pertains to the absence of a functional measure of immune competence, such as the ability to respond to viral infections or immunization. This study also did not include a characterization of monocytes, though studies on ELS and immune cell differentiation have found links between ELS and non-classical monocytes (e.g., Ref. [68]). Future research would benefit from characterizing monocytes in addition to the lymphocyte subsets and inflammation presented in the current study. Furthermore, there are procedural limitations to consider. Blood samples were collected in the morning between 6 AM and 12 PM, a broad timeframe that might introduce diurnal rhythm-related influences on peripheral lymphocytes. We note that there was no difference in the time of blood draw between PI and NA youth, so this is unlikely to have impacted our findings, while also addressing diurnal variation by controlling for the time of blood draw in all analyses.

### 4.4. Conclusions

In summary, this investigation provides evidence that early rearing in an institutional setting is linked to distinct lymphocyte profiles, independent of inflammatory markers. Even after control for covariates and inflammation and adjustment for multiple comparisons, ELS exposure remains significantly correlated with an increased proportion of effector memory helper T cells, effector memory cytotoxic T cells, senescent T cells, senescent B cells, and IgD− memory B cells, while also showing a reduced proportion of helper T cells and naive cytotoxic T cells. These findings suggest that ELS may contribute to the accumulation of lymphocytes exhibiting replicative senescence, independently of inflammation, in adolescents. If replicated in external cohorts, these findings may unveil novel avenues for comprehending the development of an inflammatory phenotype linked to heightened mortality risk following exposure to ELS. 

## Figures and Tables

**Figure 1 biomolecules-14-00262-f001:**
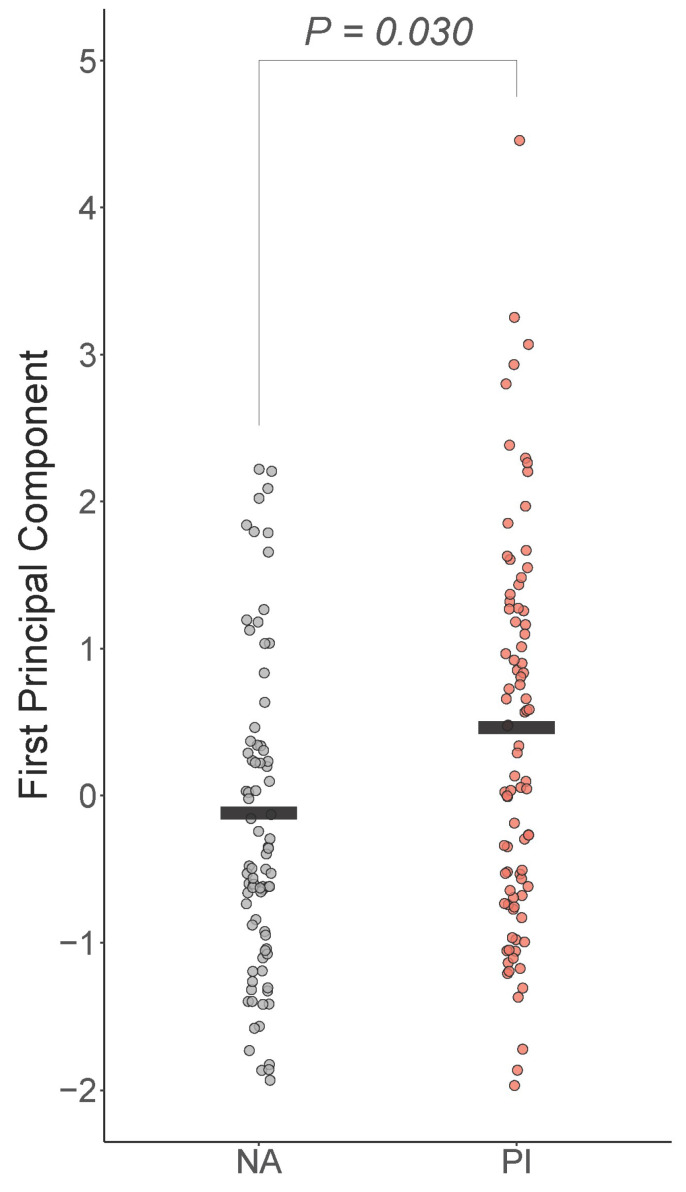
Early life adversity is positively associated with peripheral inflammation. First principal component of IL-6, CRP, and TNF-a was positively and significantly associated with early life stress, controlling for sex, age, total fat percentage, and time of blood draw. PI = previously institutionalized group, NA = non-adopted group.

**Figure 2 biomolecules-14-00262-f002:**
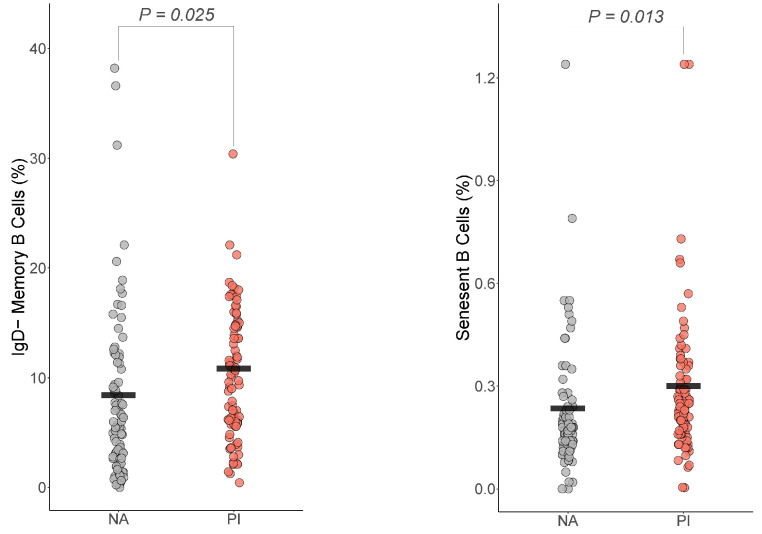
Early life adversity was associated with increased IgD− memory B cells and senescent B cells. The presented *p*-value is from the multiple regression model that controlled for sex, age, total fat percentage, time of blood draw, IL-6, and CRP for: (**a**) IgD− memory B cells, (**b**) senescent B cells. PI = previously institutionalized group, NA = non-adopted group.

**Figure 3 biomolecules-14-00262-f003:**
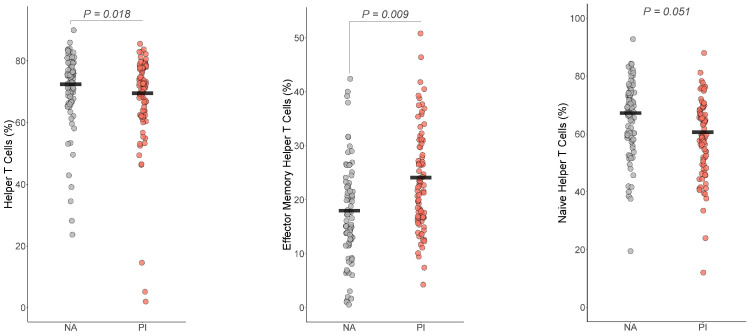
Early life stress exposure was associated with fewer Helper T cells and naïve helper T cells and increased effector memory helper T cells. The presented *p*-value is from the multiple regression model that controlled for sex, age, total fat percentage, time of blood draw, IL-6, and CRP for (**a**) helper T cells, (**b**) effector memory helper T cells, and (**c**) naïve helper T cells. PI = previously institutionalized group, NA = non-adopted group.

**Figure 4 biomolecules-14-00262-f004:**
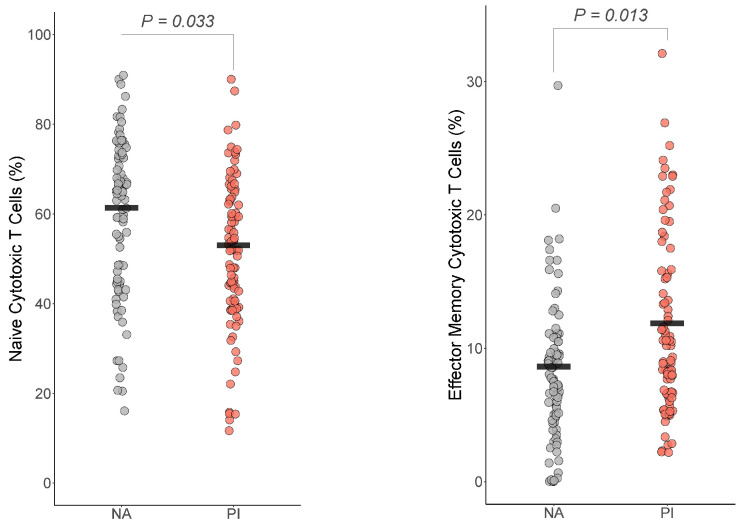
Early life stress exposure was associated with fewer naive cytotoxic T cells and effector memory cytotoxic T cells. The presented *p*-value is from the multiple regression model that controlled for sex, age, total fat percentage, time of blood draw, IL-6, and CRP for (**a**) naive cytotoxic T cells and (**b**) effector memory cytotoxic T cells. PI = previously institutionalized group, NA = non-adopted group.

**Figure 5 biomolecules-14-00262-f005:**
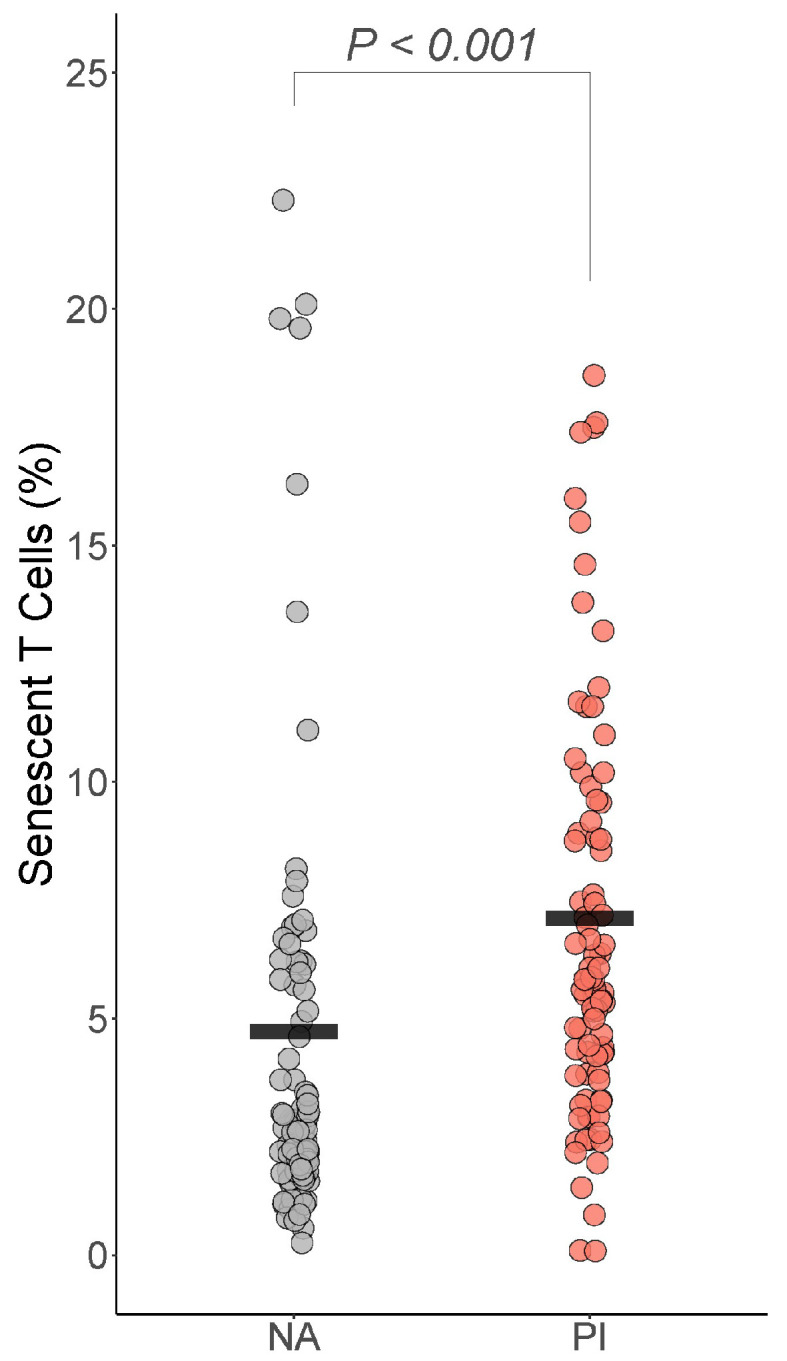
Early life stress exposure was associated with more senescent T cells. The presented *p*-value is from the multiple regression model that controlled for sex, age, total fat percentage, time of blood draw, IL-6, and CRP. PI = previously institutionalized group, NA = non-adopted group.

**Table 1 biomolecules-14-00262-t001:** Participant characteristics, Mean ± SD or N(%) where indicated.

	Non-Adopted (N = 96)	Adopted (N = 95)	Group Difference (*p*-Value of *t*-Test or chi-Square)
**Age (years)**	14.7 ± 2.32	15.9 ± 2.40	<0.001
Missing	1 (1.0%)	2 (2.1%)	
**Female**	50 (52.1%)	60 (63.2%)	n.s.
**Primary caregiver educational attainment**			n.s.
High School Degree or GED	2 (2.2%)	0 (0%)	
Some College, Community college or 2 year degree	10 (10.4%)	1 (1.1%)	
4 year degree	28 (29.2%)	28 (29.5%)	
Some graduate School or Advanced degree	53 (55.2)	65 (68.4%)	
Unreported	3 (3.1%)	1 (1.1%)	
**Total household income**			n.s.
<USD 85,000	16 (16.7%)	13 (13.7%)	
USD 85,001–USD 100,000	10 (10.4%)	8 (8.4%)	
USD 100,001–USD 150,000	22 (22.9%)	18 (18.9%)	
USD 150,001–USD 200,000	14 (14.6%)	27 (28.4%)	
>USD 200,000	29 (30.2%)	28 (29.5%)	
Missing	5 (5.2%)	1 (1.1%)	
**Child reported ethnicity**			*p* < 0.001
Hispanic or Latinx	3 (3.1%)	20 (21.1%)	
Not Hispanic or Latinx	86 (89.6%)	67 (70.5%)	
Unknown	1 (1%)	4 (4.2%)	
Missing	6 (6.3%)	4 (4.2%)	
**Adoptive Region of Origin**			–
Central America and the Caribbean	-	7 (7.4%)	
East, West, and Central Africa	-	13 (13.7%)	
Eastern Asia	-	30 (31.6%)	
Eastern Europe	-	14 (14.7%)	
South America	-	16 (16.8%)	
Southeast and South–Central Asia	-	14 (14.7%)	
Unreported	-	1 (1.1%)	
**Child reported race**			<0.001
Indigenous to the Americas	1 (1.0%)	14 (14.7%)	
Asian	3 (3.1%)	45 (47.4%)	
Black	2 (2.1%)	13 (13.7%)	
White	78 (81.3%)	15 (15.8%)	
More than 1 race	11 (11.5%)	6 (6.3%)	
Unreported	0 (0%)	1 (1.1%)	
Missing	1 (1%)	1 (1.1%)	
**Age at adoption (months)**	-	16.1 ± 11.9	-
**Months spent in institutional care**	-	14.2 ± 10.7	-
Missing	-	10 (10.5%)	-
**Body Mass Index (BMI)**	21.4 ± 4.94	22.7 ± 5.86	n.s.
Missing	1 (1.0%)	0 (0%)	
**Total Adipose Tissue (%)**	27.8 ± 8.5	29.8 ± 9.1	n.s.
Missing	1 (1%)	2 (2.1%)	
**No known allergies (self-report)**	72 (75.8%)	71 (74.3%)	n.s.
Missing	1 (1%)	0 (0%)	
**Fasted at blood draw**	88 (91.7%)	80 (84.2%)	0.044
Missing	4 (4.2%)	2 (2.1%)	
**Time of blood draw (HH:MM)**	08:39 ± 1:04	08:47 ± 1:01	n.s.
Missing	1 (1.0%)	0 (0%)	
**CMV Seropositive**	28 (29.2%)	76 (80.0%)	<0.001
Missing	9 (9.4%)	5 (5.3%)	
**IL-6 (pgml)**	1.33 ± 1.31	1.98 ± 2.08	0.013
Missing	8 (8.3%)	3 (3.2%)	
**TNF-a (pgml)**	10.2 ± 2.15	11.1 ± 7.21	n.s.
Missing	8 (8.3%)	3 (3.2%)	
**CRP (ngml)**	852 ± 1520	1440 ± 2200	0.038
Missing	8 (8.3%)	3 (3.2%)	

**Table 2 biomolecules-14-00262-t002:** Association between early life stress exposure and cell subsets.

Cell Subset	Cell Type	Est	SE	t	*p*-Value ^d^	Cohen’s $\;f^{2}$
B cells	B lymphocytes	0.69	0.97	0.71	0.567	<0.01
	IgD− memory B cells ^a^	0.33	0.12	2.72	0.025 *	0.07
	IgD+ memory B cells ^a^	−0.13	0.09	−1.52	0.178	0.01
	Naive B cells	−1.13	1.62	−0.70	0.567	0.02
	Senescent B cells ^b^	0.08	0.03	3.09	0.013 *	0.05
T cells	T cells ^c^	−3.77	2.17	−1.74	0.152	0.01
	Helper T cells	−3.72	1.29	−2.90	0.018 *	0.06
	Central memory naive T cells	0.33	0.66	0.50	0.680	0.01
	Effector memory helper T cells	4.41	1.30	3.40	0.009 *	0.19
	Effector helper T cells ^a^	0.02	0.08	0.19	0.893	<0.01
	Naive helper T cells	−4.50	1.91	−2.36	0.051 ^t^	0.10
Cytotoxic T cells	Cytotoxic T cells	2.61	1.19	2.19	0.07 ^t^	0.04
	Central memory cytotoxic T cells ^a^	0.00	0.05	0.06	0.951	<0.01
	Effector cytotoxic T cells	3.95	2.39	1.65	0.177	0.03
	Effector memory cytotoxic T cells	3.03	0.96	3.17	0.013 *	0.09
	Naive cytotoxic T cells	−6.88	2.68	−2.57	0.033 *	0.07
Regulatory T cells	Regulatory T Cells	0.94	0.62	1.52	0.178	0.02
Senescent T cells	Senescent T cells ^a^	0.46	0.09	4.94	<0.001 *	0.19
	Senescent cytotoxic T cells	3.61	2.29	1.58	0.178	0.04
	Senescent helper T cells	3.20	1.52	2.11	0.077 ^t^	0.03
Natural Killer T cells	Natural killer T cells ^a^	0.13	0.09	1.50	0.178	0.04

Notes. ^a^ Cell type was log transformed with a continuity correction of 1 added; ^b^ cell type was square root transformed; ^c^ permutation test was performed; and ^d^
*p*-value was BH corrected. ^t^ Statistical significance (*p*-value < 0.05) did not survive correction for multiple comparisons. All *t*-tests had d.f. of 160. * indicates BH corrected *p*-value < 0.05.

## Data Availability

Data will be made available upon reasonable request.

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
