# Peer review of "Early Life Stress Is Associated with Alterations in Lymphocyte Subsets Independent of Increased Inflammation in Adolescents"

_biomolecules, 2024, doi:10.3390/biom14030262_

Round 1

Reviewer 1 Report

Comments and Suggestions for Authors

The paper from Reid and collegues describes the association between early life stressful events (in this case, adoption) and alteration in lymphcytes substes in a population of teens and young adults . The paper is well written, the introduction is very rich and informative, the results are cleary presented.

I have few minor points:

Immunophenotype was performed on whole blood or on purified lymphocytes? If phenotype was performed on whole blood, why did the authors choose not to use anti CD45 antibody to do the CD45 gating?

Please provide the company from which fluorochromes conjugfated antibodies were purchase, the type of flow cytometer used and the software used to acquire and analyze the data.

Table 1, check the number of females in the non adopted group (there is a typing error)

Table 1and study population section. I am not sure that the use of the word “race” is adequate speaking of human beings.

Table 1 , I don’t understand the numbers relative to BMI and to months  spent in institutional care. In particular, while it is clear that 14,2 refers to months, I cannot figure out what 10.7 refers to (the number in brackets). Please explain numbers that do not refers to absolute number of participants or percentage of total enrolled subjects.

Page 21: please change “Tcell Regulation Cells” with “Regulatory T cells” or “Treg”

Author Response

Reviewer 1

The paper from Reid and collegues describes the association between early life stressful events (in this case, adoption) and alteration in lymphcytes substes in a population of teens and young adults . The paper is well written, the introduction is very rich and informative, the results are cleary presented.

I have few minor points:

Immunophenotype was performed on whole blood or on purified lymphocytes? If phenotype was performed on whole blood, why did the authors choose not to use anti CD45 antibody to do the CD45 gating?

Immunophenotyping was performed on whole blood samples stained after red blood cell (RBC) lysis by alkali. While we agree that it is common to use a CD45 vs SSC approach for leukemia/lymphoma flow cytometry cases, when performing lymphocyte subset analysis, is standard practice to select the lymphocytes based on forward versus side scatter (FSC vs SSC) plots in many research and clinical labs. This method shows distinct populations for unlysed RBCs, lymphocytes, monocytes, and granulocytes.

We have updated the methods section to reflect this.

Please provide the company from which fluorochromes conjugfated antibodies were purchase, the type of flow cytometer used and the software used to acquire and analyze the data.

Antibodies were purchased from either BD Biosciences (San Jose, CA) or BioLegend (San Diego, CA).  Samples were collected and analyzed on a Fortessa X20 flow cytometer (BD Biosciences, San Jose, CA).  The flow cytometric data was collected with FACSDiva (BD Biosciences, San Jose, CA), and was analyzed with FloJo (Ashland, OR).

The methods section has been updated to reflect this.

Table 1, check the number of females in the non adopted group (there is a typing error)

Thank you for catching this, this has been fixed.

Table 1 and study population section. I am not sure that the use of the word “race” is adequate speaking of human beings.

While race is not a biological construct, it is a social one that has important consequences for children's lives, and thus is included in this report. Further, reporting race and ethnicity is mandated by the US National Institutes of Health (NIH), consistent with the Inclusion of Women, Minorities, and Children policy. 

Table 1 , I don’t understand the numbers relative to BMI and to months  spent in institutional care. In particular, while it is clear that 14,2 refers to months, I cannot figure out what 10.7 refers to (the number in brackets). Please explain numbers that do not refers to absolute number of participants or percentage of total enrolled subjects.

Those numbers refer to the standard deviation. However, we agree that this could be clearer and have updated Table 1 to indicate N (%) and Mean ± SD where applicable for clarity.

Page 21: please change “Tcell Regulation Cells” with “Regulatory T cells” or “Treg”

Thank you for this suggestion, the has been updated to Regulatory T cells

Reviewer 2 Report

Comments and Suggestions for Authors

Thank you for the invitation to review this interesting text.

The content fully corresponds to the topic, the article is well-written and gives a clear picture of the actual state of knowledge. The literature in this area was thoroughly reviewed.

I have a few minor comments/suggestions.

Line 175: ‘could not forgo’- letter missing in word

Line 187: double dots

In my opinion, Table 1 (the characteristics of the participants) should also include the results of anthropometric studies DEXA (fat %) or at least BMI. As we know, obesity is also associated with chronic low-grade inflammation.

I had a few more questions, but I found the answer to most of them in the ‘Limitations’ (4.3 section). When considering the relationship between immune status in children and adolescents and environmental factors, we cannot overlook factors (maternal?) interacting in fetal life (metabolic programming), which can certainly complete the overall picture, influence the conclusion or completely change it.

In addition, these results are not linked to the clinical picture of the participants, such as the prevalence of allergies, autoimmune or other chronic diseases, which would greatly enrich the perception of these observations and confirm (?) the conclusion made.

 I believe that the article will attract a wide range of readers

Author Response

Reviewer 2

Thank you for the invitation to review this interesting text. 

The content fully corresponds to the topic, the article is well-written and gives a clear picture of the actual state of knowledge. The literature in this area was thoroughly reviewed. 

Thank you so much!

I have a few minor comments/suggestions.

Line 175: ‘could not forgo’- letter missing in word

We have changed “not forgo” to “could not go without medications” for clarity

Line 187: double dots

Thank you, this has been fixed.

In my opinion, Table 1 (the characteristics of the participants) should also include the results of anthropometric studies DEXA (fat %) or at least BMI. As we know, obesity is also associated with chronic low-grade inflammation.

We agree, and the percent body fat is included in Table 1. Additionally, we have added BMI to table 1. All analyses further control for % adipose tissue (body fat %) as determined by DEXA.

I had a few more questions, but I found the answer to most of them in the ‘Limitations’ (4.3 section). When considering the relationship between immune status in children and adolescents and environmental factors, we cannot overlook factors (maternal?) interacting in fetal life (metabolic programming), which can certainly complete the overall picture, influence the conclusion or completely change it.

We agree, and have added this important point to the limitations section, to read as follows: 

“Maternal and fetal factors involved in metabolic programming may explain or influence the conclusions drawn from the current results.”

In addition, these results are not linked to the clinical picture of the participants, such as the prevalence of allergies, autoimmune or other chronic diseases, which would greatly enrich the perception of these observations and confirm (?) the conclusion made.

The participants were generally healthy, given the exclusion criteria noted in the methods section:

Youth participants were excluded if the parents reported that youth born prematurely (less than 37 weeks of the gestational age), had congenital and/or chromosomal disorders (e.g., cerebral palsy, FAS, intellectual disability, Turner Syndrome, Down Syndrome, Fragile X), autism spectrum disorders, history of serious medical illness (e.g., cancer, organ transplant), if they were taking systemic glucocorticoids, or if they were taking beta-adrenergic medications and could not forgo medication for 24-48 hours before testing. 

Further, ~75% of participants from both groups had no known allergies, and this is now reported in Table 1.

We have added the following to the discussion section for context: “The cohort studied consists of clinically healthy adolescents, with three-quarters of the sample reporting no known allergies.”

Reviewer 3 Report

Comments and Suggestions for Authors

ABSTRACT must be structured and divided on 4 sections as follows:

-Introduction- aim of the study must be indicated mandatory, other data are not necessary in the abstract. SO Indicate clearly, what the goal of your study was.

Methods-besides the list of patients, you must list out methods performed for the study. Statistical analysis should be omitted from the abstract List out all methods performed in the manuscript ieCRP and… were determined by ELISA, CMV with chemiluminescent immunoassay, immunophenotyping ….

-Results – represent the most important results numerically including P-values.

-Conclusions- According to your aim answer what was your conclusion, as already stated in the penultimate phrase of the abstract. Last phase have to be omitted from the abstract because this is not conclusion (Novel insights and future replication are not the answer to the aim.

MATEIAL AND METHODS

Statistics must be presented in detailed as follows

-          Declare about testing distribution normality

-          Declare how you expressed that you measured. With mean and SD or MEDIAN and range for continuous data? Declare that categorical data were expressed with absolute number and percentage

-          Declare about performing parametric and or non-parametric statistics

-          It is important to know which type of data enter to regression analysis.

-          Why did you perform regression analysis without previous correlation analysis? Only parameters that showed correlation, could be included to the regression model.

RESULTS

Figures and tables must be self-explanatory. Title must strongly indicate what you represented at the figure or table. Legend shod be included with all data (statistical test, explained axis, explained abbreviation. Do not comment the results bellow the figure. Comment should be in the main text of the results section.

ABSTRACT must be structured and divided on 4 sections as follows:

-Introduction- aim of the study must be indicated mandatory, other data are not necessary in the abstract. SO Indicate clearly, what the goal of your study was.

Methods-besides the list of patients, you must list out methods performed for the study. Statistical analysis should be omitted from the abstract List out all methods performed in the manuscript ie CRP and… were determined by ELISA, CMV with chemiluminescent immunoassay, immunophenotyping ….

-Results – represent the most important results numerically including P-values.

-Conclusions- According to your aim answer what was your conclusion, as already stated in the penultimate phrase of the abstract. Last phase have to be omitted from the abstract because this is not conclusion (Novel insights and future replication are not the answer to the aim.

MATEIAL AND METHODS

Statistics must be presented in detailed as follows

-          Declare about testing distribution normality

-          Declare how you expressed that you measured. With mean and SD or MEDIAN and range for continuous data? Declare that categorical data were expressed with absolute number and percentage

-          Declare about performing parametric and or non-parametric statistics

-          It is important to know which type of data enter to regression analysis.

-          Why did you perform regression analysis without previous correlation analysis? Only parameters that showed correlation, could be included to the regression model.

RESULTS

Figures and tables must be self-explanatory. Title must strongly indicate what you represented at the figure or table. Legend shod be included with all data (statistical test, explained axis, explained abbreviation. Do not comment the results bellow the figure. Comment should be in the main text of the results section.

Author Response

Reviewer 3

ABSTRACT must be structured and divided on 4 sections as follows: 

-Introduction- aim of the study must be indicated mandatory, other data are not necessary in the abstract. SO Indicate clearly, what the goal of your study was. 

Methods-besides the list of patients, you must list out methods performed for the study. Statistical analysis should be omitted from the abstract List out all methods performed in the manuscript ieCRP and... were determined by ELISA, CMV with chemiluminescent immunoassay, immunophenotyping ....

-Results – represent the most important results numerically including P-values.

-Conclusions- According to your aim answer what was your conclusion, as already stated in the penultimate phrase of the abstract. Last phase have to be omitted from the abstract because this is not conclusion (Novel insights and future replication are not the answer to the aim.

Thank you for these suggestions. We have revised the abstract, with the exception of the headed sections and numerical results with p-values as the word limit is already tight in the abstract and we are following the format as described in the guide to authors:

Abstract: The abstract should be a total of about 200 words maximum. The abstract should be a single paragraph and should follow the style of structured abstracts, but without headings: 1) Background: Place the question addressed in a broad context and highlight the purpose of the study; 2) Methods: Describe briefly the main methods or treatments applied. Include any relevant preregistration numbers, and species and strains of any animals used; 3) Results: Summarize the article's main findings; and 4) Conclusion: Indicate the main conclusions or interpretations. The abstract should be an objective representation of the article: it must not contain results which are not presented and substantiated in the main text and should not exaggerate the main conclusions.”

The new abstract reads as follows:

Early life stress (ELS) is linked to an elevated risk of poor health and early mortality, with emerging evidence pointing to the pivotal role of the immune system in long-term health outcomes. While recent research has focused on the impact of ELS on inflammation, this study examined the impact of ELS on immune function, including CMV seropositivity, inflammatory cytokines and lymphocyte cell subsets in an adolescent cohort. This study used data from the Early Life Stress and Cardiometabolic Health in Adolescence Study (N=191, aged 12 to 21 years, N=95 exposed to ELS). We employed multiple regression to investigate the association between ELS, characterized by early institutional care, cytomegalovirus (CMV) seropositivity (determined by chemiluminescent immunoassay), inflammation (CRP, IL-6 and TNF-a determined by ELISA), and 21 immune cell subsets characterized with flow cytometry (16 T cell subsets and 5 B cell subsets). Results reveal a significant association between ELS and lymphocytes that was independent of the association between ELS and inflammation: ELS was associated with increased effector memory helper T cells, effector memory cytotoxic T cells, senescent T cells, senescent B cells, and IgD- memory B cells compared to non-adopted youth. ELS was also associated with reduced percentages of helper T cells and naive cytotoxic T cells. Exploratory analyses found that the association between ELS and fewer helper T cells and increased cytotoxic T cells remained even in cytomegalovirus (CMV) seronegative youth. These findings suggest that ELS is associated with cell subsets that are linked to early mortality risk in older populations and markers of replicative senescence, separate from inflammation, in adolescents.

MATERIAL AND METHODS

Statistics must be presented in detailed as follows:

- Declare about testing distribution normality

We examined Q-Q plots to assess normality and considered transformations of the response variables that best met normality and homogeneity of variance. We did not perform formal tests of normality. This has been added to the methods section.

Declare about performing parametric and or non-parametric statistics

Please see page 7 where we indicated that we performed a permutation test for the T-cell subset as we were not able to find an adequate transformation that met homogeneity of variance and normality

Why did you perform regression analysis without previous correlation analysis? Only parameters that showed correlation, could be included to the regression model

We believe the reviewer here is referring to variables, not parameters. In particular, the control variables (covariates). Covariates were selected based on theory, not whether they were found to be statistically significant within this sample. While it is true that a lack of significance in our sample would indicate these variables can not be confounding variables in the sample, that does not mean they may not be confounders in the population.

Declare how you expressed that you measured. With mean and SD or MEDIAN and range for continuous data? Declare that categorical data were expressed with absolute number and percentage

We agree that this could be clearer and have updated Table 1 to indicate N (%) and Mean ± SD where applicable for clarity.

RESULTS

Figures and tables must be self-explanatory. Title must strongly indicate what you represented at the figure or table. Legend should be included with all data (statistical test, explained axis, explained abbreviation. Do not comment the results bellow the figure.Comment should be in the main text of the results section.

Thank you for this suggestion. We have updated the title and text for the figures.